# Integrating Heuristic Methods with Deep Reinforcement Learning for Online 3D Bin-Packing Optimization

**DOI:** 10.3390/s24165370

**Published:** 2024-08-20

**Authors:** Ching-Chang Wong, Tai-Ting Tsai, Can-Kun Ou

**Affiliations:** Department of Electrical and Computer Engineering, Tamkang University, New Taipei City 25137, Taiwan; 609470124@gms.tku.edu.tw (T.-T.T.); 611470070@o365.tku.edu.tw (C.-K.O.)

**Keywords:** 3D bin-packing, deep reinforcement learning, proximal policy optimization, heuristic algorithms

## Abstract

This study proposes a method named Hybrid Heuristic Proximal Policy Optimization (HHPPO) to implement online 3D bin-packing tasks. Some heuristic algorithms for bin-packing and the Proximal Policy Optimization (PPO) algorithm of deep reinforcement learning are integrated to implement this method. In the heuristic algorithms for bin-packing, an extreme point priority sorting method is proposed to sort the generated extreme points according to their waste spaces to improve space utilization. In addition, a 3D grid representation of the space status of the container is used, and some partial support constraints are proposed to increase the possibilities for stacking objects and enhance overall space utilization. In the PPO algorithm, some heuristic algorithms are integrated, and the reward function and the action space of the policy network are designed so that the proposed method can effectively complete the online 3D bin-packing task. Some experimental results illustrate that the proposed method has good results in achieving online 3D bin-packing tasks in some simulation environments. In addition, an environment with image vision is constructed to show that the proposed method indeed enables an actual robot manipulator to successfully and effectively complete the bin-packing task in a real environment.

## 1. Introduction

The three-dimensional bin-packing (3D-BP) problem is a classic combinatorial optimization problem aimed at arranging objects within container boxes to maximize space utilization [1,2]. Historically, the manual arrangement of items for packing has been labor-intensive and often yields suboptimal packing configurations. The 3D-BP problem is widely regarded as an NP-hard problem and can be categorized into offline and online. In the offline 3D-BP problem, the numbers and sizes of objects are known in advance, while the online 3D-BP problem is more challenging as it requires packing objects neatly into containers without prior knowledge of their quantities or sizes.

In the realm of packing problems, the one-dimensional bin-packing (1D-BP) problem stands out as a typical combinatorial optimization challenge, aiming to fit numerous items into the fewest number of boxes with predetermined capacities [3,4]. A review of relevant algorithms addressing the 1D-BP problem over the past two decades reveals its complexity and provides insights into its solution [5]. As dimensions increase, from two dimensions (2D) to three dimensions (3D), packing complexity escalates accordingly.

Early studies on 3D packing tasks primarily relied on well-designed heuristic algorithms [2,6,7,8], often inspired by manual packing experiences. For instance, research [9] introduced the layer-based approach three-dimensional bin-packing problem, dividing boxes into layers along two possible directions to maximize item arrangements. Expanding upon this approach, research [10] has proposed a three-stage layer-based heuristic to tackle the three-dimensional single-bin-size bin-packing problem. Additionally, research [11] presented a multi-objective 3D packing algorithm optimizing container usage and balancing weights across containers. The research in [12] combined research [9,11] to devise a multi-stage optimization algorithm for solving the 3D-BP problem.

Despite the effectiveness of heuristic algorithms, inherent limitations exist, such as lack of learning capabilities [13] and exponential growth in solution time with increasing problem sizes [14], rendering reliance solely on heuristic algorithms impractical for achieving optimal packing. Recently, deep reinforcement learning (DRL) has demonstrated promising performance across various domains, prompting its application in 3D packing problems [15,16,17,18,19,20]. Leveraging DRL enables continuous learning from manual packing experiences, aligning packing outcomes with human preferences. Particularly, addressing online 3D-BP using DRL has become a recent research trend [21,22]. Methods such as model-based reinforcement learning proposed by the research in [23], based on research [24], optimize container space utilization through DRL. Effective and easily implementable constrained DRL methods within the actor–critic framework were introduced by research [25]. Moreover, research [26] proposed an adjustable robust reinforcement learning approach to tackle online 3D-BP, effectively adjusting robustness weights to achieve an ideal balance of policy performance across various environmental conditions.

Given the NP-hard nature of the 3D-BP problem and the limitations of heuristic algorithms, there is a clear research gap in developing effective and efficient methods for online 3D-BP. This study aims to address this gap by combining reasonable constraints and heuristic algorithms with DRL to improve space utilization in practical scenarios involving robot manipulators for packing. These constraints include placement direction constraints and partial support constraints, which are necessary due to the limitations of robot manipulators without specially designed end effectors. The placement direction constraints prohibit the rotation of objects along the x and y axes, while partial support constraints increase the possibilities for stacking objects and enhance overall container space utilization.

This study is organized into four sections. Section 1 introduces the background and research objectives. Section 2 details the methodology for solving the 3D-BP problem using heuristic algorithms in conjunction with a DRL system. Section 3 describes the experimental environment established for the online 3D-BP task, along with the presentation of simulation and actual experimental results. Finally, Section 4 provides the conclusions and future work.

## 2. Methodology

This study adopts the classic and mainstream extreme point heuristic algorithm [27] within the realm of heuristic algorithms and enhances it with custom-designed modifications tailored to the overall online 3D-BP tasks.

### 2.1. Extreme Point Sorting Method

In the packing task, the current placement point of each item utilizes the extreme point method proposed by research [27], where each item’s stacking generates three extreme points: upper, left-front, and right-rear. Placing items at the upper extreme point may result in residual space due to varying item sizes. This residual space cannot accommodate all items, leading to space wastage. To minimize space wastage, we devise the extreme point priority sorting method to place items in positions with minimal wasted space. The operation flow is shown in Figure 1.

The method calculates the residual space generated after placing the current item at all placement points. If the residual space cannot accommodate all previously placed items, it is considered wasted space. If the residual space can accommodate one item, it is considered available space. All placement points are sorted based on wasted space, with those producing less wasted space prioritized. The prioritization sequence for placement points is as follows: points with no residual space, points with available space, and finally, points with wasted space, arranged from lowest to highest wasted space. Algorithm 1 provides the pseudocode for the extreme point priority sorting method.
**Algorithm 1**: Extreme Point Priority Sorting MethodInput EP: List of Extreme PointsInput I: Item to be added to the 3D binInput K: List of items already in the 3D bin**function** Calculate_remain_space(I, EP):  **return** space remaining**function** Dec_waste_space(rem_space*_i_*):  **if** space remaining < for all object ∈ K    **return** *True***function** Max_score(EP, score_list):  **return** EP list ← score from high to low**for** all *i* ∈ EP **do**  rem_space*_i_* = Cauculate_remain_space(I, *i*)  **if** Dec_waste_space(rem_space*_i_*) then    score_list ← - rem_space*_i_*  **else**    score_list ← 0  **end if****end for****return** Max_score(EP, score_list)

When placing objects of different sizes, L-shaped residual spaces can be formed. Before calculating the residual space, the space generated after placement is divided into two residual spaces: one along the x-axis and one along the y-axis. The space division method follows the approach mentioned in the research of [5]. The outermost vertices of the object cut the residual space along the x-axis and y-axis. To utilize the space more efficiently, the direction with the longer residual axis is given a larger space. The overlapping space is defined as transferable space, which is allocated to the residual space along the longer axis.

### 2.2. Packing Constraints

Packing constraints are rules derived from manual packing experiences, designed to optimize the packing task. This study uses container space state representation to implement the constraints of the heuristic algorithm and packing constraints, including a simple non-complete support constraint.

#### 2.2.1. Container Space State Representation

To enable the heuristic algorithm and related constraints to function effectively in the packing task, the state of the container space must first be represented. There are various methods for handling container space. For instance, the method used by study [28] involves cutting the remaining container space after placing an item and using an algorithm to determine whether the item can be placed to fit the remaining space. In contrast, study [25] proposed creating a 2D height map for container space projection, where the values filled in the 2D map represent the current height. However, the height map overlooks three-dimensional information, potentially causing discrepancies in some situations. To address this, study [13] proposed using a 3D grid to represent three different space states, wasted space, available space, and unavailable space, using the 3D grid as input to neural networks.

Building on the method by study [13], this study proposes creating a 3D grid to represent container space states, but with consideration for the support state of the space. The container space is divided into grids with a volume of 1, where each grid’s value indicates the current space state. The three defined space states, as shown in Figure 2, are space occupied, empty space without item support, and empty space with item support. This container space state representation allows for the evaluation of the rationality of item stacking within the container.

#### 2.2.2. Partial Support Constraints

In traditional methods for stacking objects, complete support constraints are used to maintain static stability. This requires that the bottom of the object be fully supported by other objects or the container itself, prohibiting any overhanging placements. However, complete support constraints overly restrict space utilization and limit the heuristic algorithm’s flexibility.

In this study, we introduce partial support constraints to allow for more effective space use while maintaining stability. This is achieved through the object’s area and the center of the bottom of the object, with three main constraints: half the length of the object *L_object_* must be less than the length of the placement space *L_space_*, half the width of the object *W_object_* must be less than the width of the placement space *W_space_*, and the position directly below the center of the object *object_pos_* must be supported by another object or the container. These constraints are mathematically represented by
(1)$$\frac{1}{2}L_{object}<L_{space}$$
(2)$$\frac{1}{2}W_{object}<W_{space}$$
(3)$${object}_{pos}=(x_{center},{\;y}_{center},\;{\;z}_{center})$$

By adhering to these three constraints, objects achieve partial support, ensuring stability and preventing collapse due to inadequate support. Combining the extreme point method with partial support constraints increases the possibilities for item placement, thereby enhancing space utilization.

### 2.3. Integration of the Heuristic Algorithm with Deep Reinforcement Learning

In online 3D-BP tasks, training solely with DRL can result in poor training outcomes due to significant discrepancies in the definition of strategy quality. Therefore, heuristic algorithms are typically integrated into the training process. This section explains how heuristic algorithms and constraints are integrated into DRL, divided into three parts: extreme points and extreme point priority sorting constraints, deepest bottom left with fill, and reward function design.

#### 2.3.1. Extreme Points and Extreme Point Priority Sorting Constraints

This study integrates extreme points and the extreme point priority sorting method into the action strategy of DRL. When placing objects into the target container, the current extreme point is identified using the extreme point algorithm, which becomes the chosen placement for the action strategy. As illustrated in Figure 3, the currently placed object generates three extreme points: upper, right, and front. The extreme points generated by previously placed objects are categorized as other extreme points. The upper extreme point is classified as another extreme point in the strategy selection. Thus, the action strategy offers three placement positions and two placement directions. Among the other extreme points, many placement points exist. Using the designed extreme point priority sorting method, the wasted space caused by each placement is calculated and sorted, arranging the extreme points from least to most wasted space. To determine whether an object can be placed in the container, the 3D grid state of the container is used to prevent unreasonable placement scenarios.

#### 2.3.2. Deepest Bottom Left with Fill

The Deepest Bottom Left with Fill (DBLF) heuristic algorithm [29] aims to place objects as close to the target container’s corners and edges as possible. Building on the method by the research in [13], this study integrates DBLF into deep reinforcement learning by creating a 3D grid-based placement score space, similar to the 3D grid state method for container space. The container space is divided into grids with a volume of 1, where each grid’s value represents the placement score of an object.

Initially, the space score values are set to decrease from the bottom-left corner to the top-right corner, as illustrated in Figure 4. The figure shows a 2D score distribution, and each layer follows the depicted distribution. Stacking these distributions forms the initial 3D score space. The scoring is primarily based on whether there are objects near the space. As shown in Figure 5, if a space is already occupied, its grid score is removed; if a space is unoccupied but has neighboring objects, its grid score is increased by 3 points; if a space is unoccupied and has no neighboring objects, its grid score remains unchanged.

The reward function is designed to encourage the placement of objects in higher-scoring spaces, achieving the DBLF algorithm’s effect. This method ensures that objects are placed efficiently, maximizing space utilization by guiding the placement towards optimal positions within the container.

#### 2.3.3. Reward Function Design

This study uses Proximal Policy Optimization (PPO) [30] combined with heuristic algorithms to train the online 3D-BP task. The reward function, a critical component in integrating heuristic algorithms, is primarily based on the placement score of objects. The reward score is derived from the placement space score, supplemented by the wasted space generated after placing the object. The less wasted space produced, the higher the reward score.

The feasibility of the chosen placement position and direction also affects the reward score. If an object cannot be placed, a penalty is applied by deducting an appropriate amount from the reward score. The final total reward score considers the current space utilization rate for overall scoring. The reward function in the online 3D-BP environment uses a proportional addition format. When an object cannot be placed, it signifies the end of an episode. The total reward for that episode, $R_{total}$, can be expressed by
(4)$$R_{total}=\gamma \cdot \frac{\sum_{i=0}^{n}S_{i}}{n}+\beta {\cdot waste}_{first}+\beta \cdot \left(\sigma -S_{ratio}\right)$$
where $S_{i}$ is the score obtained from placing the *i* object, and *n* is the total number of objects placed in that episode. The first term represents the average placement score for the episode. ${waste}_{first}$ is the area of the unused space at the bottom of the container at the end of the episode. *σ* is a standard for space utilization, which is not a standard deviation but an average value of the standard measure for space utilization, adjusted after running the bin-packing task multiple times. The exact number of runs can vary. $S_{ratio}$ is the overall space utilization ratio for the episode, which can be expressed by
(5)$$S_{ratio}=\frac{\sum_{i=0}^{n}v_{i}}{L\cdot W\cdot H}\cdot 100$$
where $v_{i}$ is the volume of the current object, and *L*, *W*, *H* are the length, width, and height of the container, respectively. The third term represents the difference between the space utilization and the standard value. *β* and *γ* are constant proportions that must satisfy the condition 2β+γ=1. In Equation (4), $S_{i}$ refers to the score obtained from placing the *i* object, which is also the reward $R_{step}$ for each placement step. The formula can be expressed by
(6)$$R_{step}=S_{i}=\left\{\begin{array}{c} \;\rho \cdot S_{\mathrm{score}}\left({pos}_{i},\;{ori}_{i}\right)-{\mu \cdot waste}_{i},\;\;if\;{action}_{i}=0\;or\;{action}_{i}=1\;\\ \;\omega \cdot S_{i-1}{+\tau \cdot S}_{\mathrm{score}}\left({pos}_{i},\;{ori}_{i}\right),\;\;\;\;\;\;\;\;if\;{action}_{i}=2\;\; \end{array}\right.$$
where $S_{\mathrm{score}}$ refers to the grid score, which is the sum of the scores of the grids occupied by the placed object. pos*i* represents the position of the *i* placement, ${ori}_{i}$
*i* denotes the orientation of the *i* placement, and ${waste}_{i}$ is the wasted space generated during the *i* object placement. $S_{i-1}$ is the score obtained from the previous placement, and ${action}_{i}$ is the action taken for the *i* placement. The constants ρ, μ, τ, and ω are predefined ratios. When ${action}_{i}=0,\;1$, it indicates that the action strategy selects the front extreme point or the right extreme point. In these cases, $S_{i}$ is calculated by subtracting the ratio of wasted space from the placement score. When ${action}_{i}=2$, it indicates that the action strategy selects other extreme points, which have been filtered through the extreme point priority sorting method. These selected points are generally the best current positions. To prevent the heuristic algorithm from overly interfering with the learning of the action strategy, *S**i* is set to the sum of the placement score from the *i*−1 placement and a proportion of the grid score from the current placement. The reward parameters used in the function are listed in Table 1.

During the bin-packing process, two situations may arise: (i) The currently chosen placement position is not feasible, while other positions are feasible. (ii) The currently chosen orientation is not feasible, while another orientation is feasible. For these situations, two penalty mechanisms are designed to deduct placement scores and described by
(7)$$S_{i}=\left\{\begin{array}{c} S_{i}-10,\;if\;D_{stack}\left({pos}_{i},\;{ori}_{i}\right)=False\\ {\;\;\;\;\;\;and\;D}_{stack}\left({pos}_{n},\;{ori}_{i}\right)=True\\ S_{i}-5,\;if\;D_{stack}\left({pos}_{i},\;{ori}_{i}\right)=False\\ {\;\;\;\;\;\;and\;(D}_{stack}\left({pos}_{i},\;{ori}_{n}\right)=True\\ {\;\;\;\;\;\;\;or\;D}_{stack}\left({pos}_{n},\;{ori}_{n}\right)=True) \end{array}\right.$$
where $D_{stack}$ is used to determine if the placement is feasible through the grid state $S_{state}$. If the grid states occupied by the placement position comply with the constraints, the $D_{stack}$ outputs *True*; otherwise, it outputs *False*. The variables ${pos}_{n}$ and ${ori}_{n}$ represent the new placement position and orientation, respectively. If either of the two situations occur, the episode continues, but an appropriate placement score $S_{i}$ is deducted as a penalty for making an incorrect choice. By integrating these penalties and constraints into the reward function, the deep reinforcement learning model is guided to improve its decision-making process, leading to more efficient and optimal packing solutions.

## 3. Results and Discussion

### 3.1. System Architecture and Experimental Environment

The overall system architecture of this study is shown in Figure 6. In the motion control section of the robot manipulator, the MoveIt! motion planning module is utilized to plan the movements of the robot manipulator. This ensures that the manipulator can avoid all obstacles in its path and move quickly and accurately to the target position. In the visual assistance part, RGB-D images from a depth camera are used to obtain the three-dimensional length information and position of the target object. The obtained target object position is then converted into coordinates so that the robot manipulator can reach the target object to retrieve it. The information about the target object obtained is one of the inputs for the deep reinforcement learning system. Finally, the Robot Operating System (ROS) is used to transmit information between various modules and components, allowing the robot manipulator to complete the online 3D packing task. An actual experimental environment of this study is shown in Figure 7. The depth camera is mounted above the inspection area on the workbench to obtain the position and information of the target objects. The robot manipulator is installed on the right side of the workbench, with its end effector being a suction cup used for picking up target objects. 

### 3.2. Result of Model Training 

The model is based on the one provided by research [31], which shares similar constraints with this online 3D-BP task, considering only two orientations for the current object, ori(0, 0, 0) and ori(0, 0, 90). The dimensions of the objects and the container box are shown in Table 2.

To evaluate the proposed method, 100 test runs were conducted using the same model. The results are illustrated in Figure 8. Figure 8a,b show the packing results with space utilization rates of 90.08% and 90.06%, respectively. Figure 8c depicts the space utilization rate for each round of the 100 tests The proposed method achieved an average space utilization rate of approximately 83%, with the highest utilization reaching about 92% and the lowest around 74%. Figure 8d shows the number of objects placed in each of the 100 test tests. The average number of objects placed per round is around 455, with the maximum being 510 and the minimum being 405.

To compare the proposed HHPPO method with the model provided by research [31], Table 3 shows an improvement in the same model, with an average space utilization increase of 3%. The top 5% space utilization increased by up to 6%, and the bottom 5% space utilization improved by 3.4%. Additionally, Table 4 shows that the average number of objects placed increased by approximately 15. The test results confirm the effectiveness of the proposed method.

We also performed predictions on Model 2. The dimensions of the objects and the container box are shown in Table 5. We conducted 100 tests, and the results are illustrated in Figure 9. Figure 9a,b show the object placement results for a space utilization rate of 85.07%, from top and bottom views, respectively. Figure 9c presents the space utilization rate for each round of the 100 tests. It shows that the proposed method achieves an average space utilization rate of approximately 65% for Model 2. The highest space utilization rate is around 80%, and the lowest is about 48%. Figure 9d displays the number of objects placed in each of the 100 tests. The average number of objects placed per round is approximately 80, with the maximum being 105 and the minimum being 63. The result presents more challenges in achieving high space utilization compared to Model 1. This is because the sizes of all objects are closer to the size of the target container, making it difficult to completely fill the container. Moreover, the varying lengths and widths of the objects add to the complexity of stacking them neatly.

### 3.3. Experimental Results of Model Implement

The information on the objects used in the experiment is shown in Table 6. The experiments in this study are divided into two parts: online 3D-BP tasks simulated in Gazebo and tasks executed by a real robot in a real environment. Figure 10 shows the online 3D-BP task in the Gazebo simulation. Figure 11 depicts the storyboard of the online 3D-BP task in the real environment, and Figure 12 shows the front and rear views of the completed online 3D-BP task.

## 4. Conclusions

This study proposes the use of a PPO algorithm combined with heuristic algorithms to achieve online 3D-BP tasks. By leveraging visual assistance, the task can be completed in simulated and real-world environments using an actual robot manipulator. The main contributions comprise two parts: (i) In the heuristic algorithm for bin-packing, the proposed extreme point priority sorting method enhances space utilization by ranking the extreme points generated by the extreme point method based on wasted space. A 3D grid representation of container space is introduced, along with some partial support constraints designed according to the methods outlined in this study. (ii) In the deep reinforcement learning integration, a method combining heuristic algorithms with deep reinforcement learning is presented. The PPO algorithm is used, incorporating reward function design and the action space of the policy network. This combination integrates the extreme point method, DBLF algorithm, and the designed extreme point priority sorting heuristic algorithms, demonstrating promising results in the experiments. These contributions collectively advance the effectiveness and efficiency of the online 3D-BP task, providing an approach that can be applied in both simulated and real-world environments. In future work, unpacking algorithms will be used to create a more robust and efficient method to achieve efficient online 3D-BP tasks in dynamic environments. The objectives are to improve space utilization by minimizing the gaps between boxes through an advanced unpacking algorithm.

## Figures and Tables

**Figure 1 sensors-24-05370-f001:**
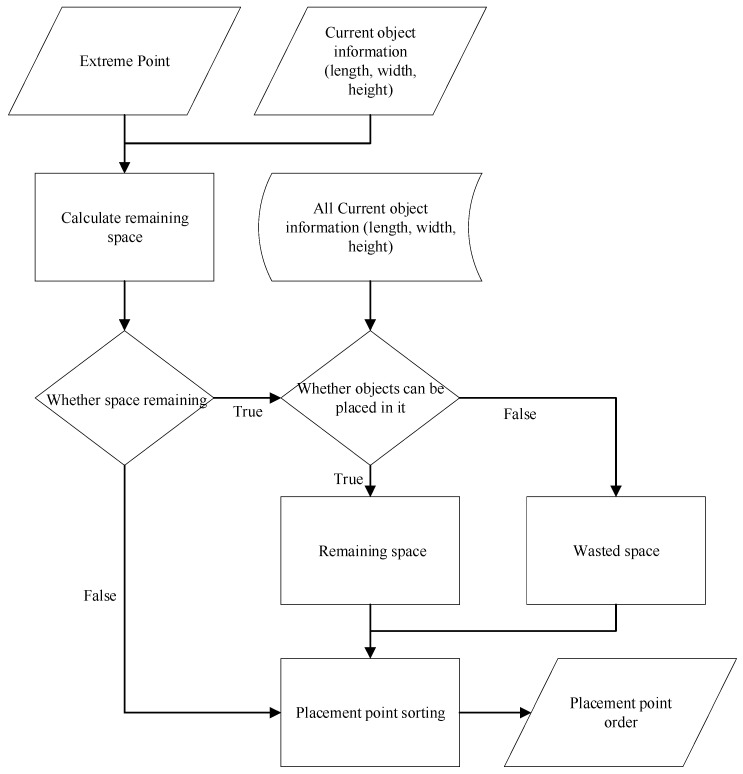
Flowchart of the extreme points priority sorted.

**Figure 2 sensors-24-05370-f002:**
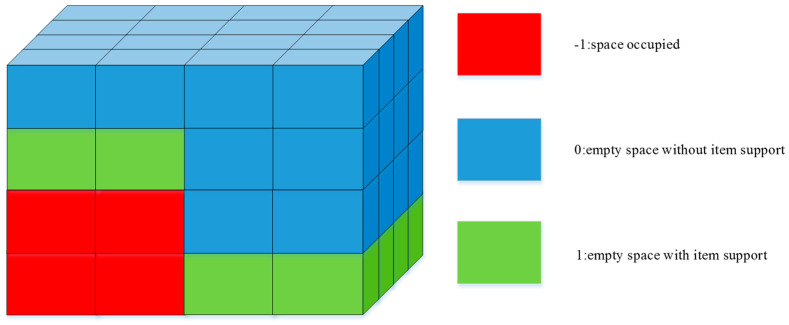
Three-dimensional grid state diagram.

**Figure 3 sensors-24-05370-f003:**
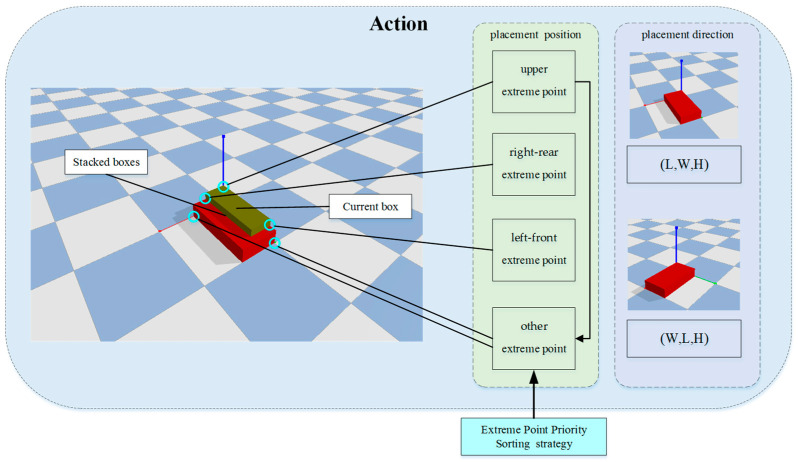
Action strategy diagram.

**Figure 4 sensors-24-05370-f004:**
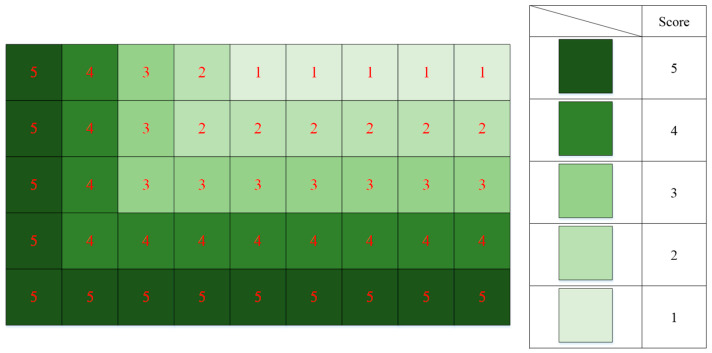
Diagram of initial space score.

**Figure 5 sensors-24-05370-f005:**
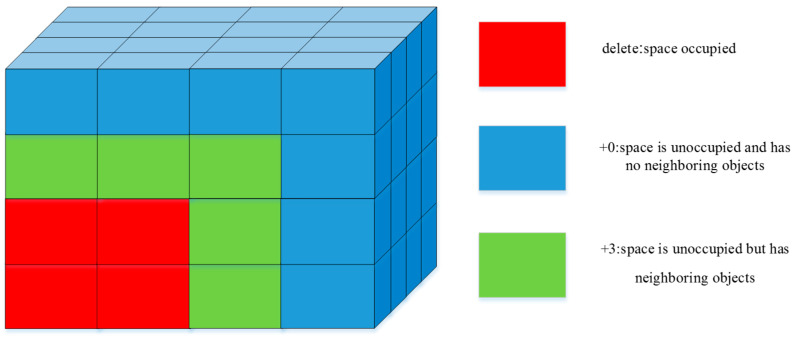
Diagram of space score.

**Figure 6 sensors-24-05370-f006:**
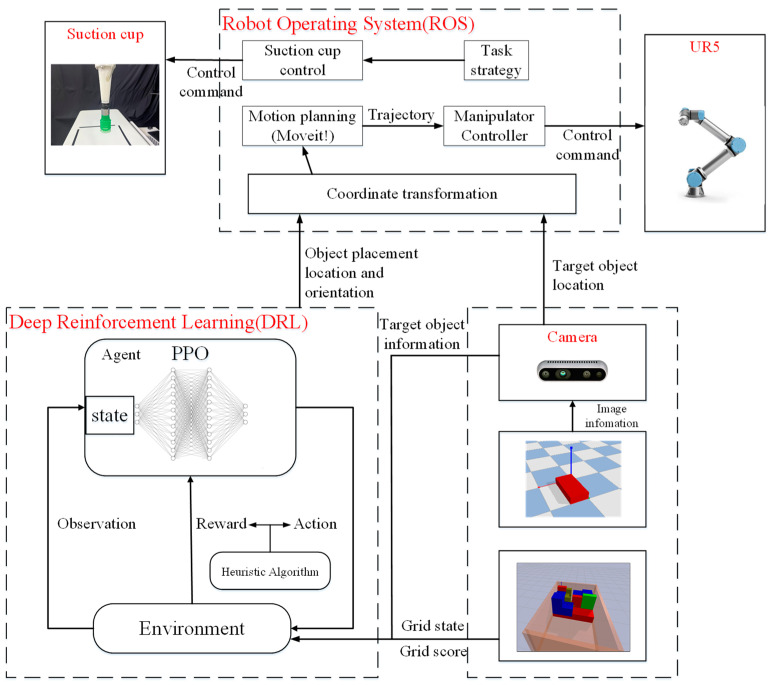
System architecture of the proposed method for 3D-BP tasks.

**Figure 7 sensors-24-05370-f007:**
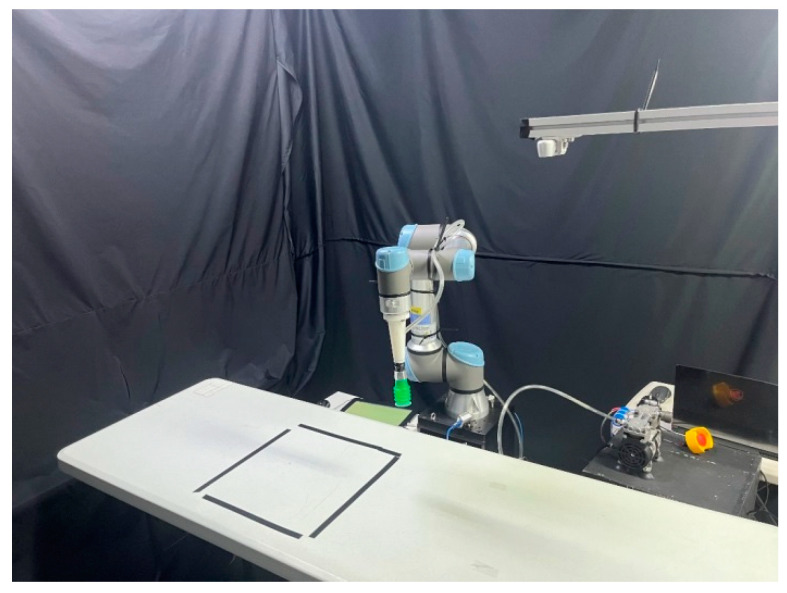
Experimental environment of a real 3D-BP task using a robot manipulator.

**Figure 8 sensors-24-05370-f008:**
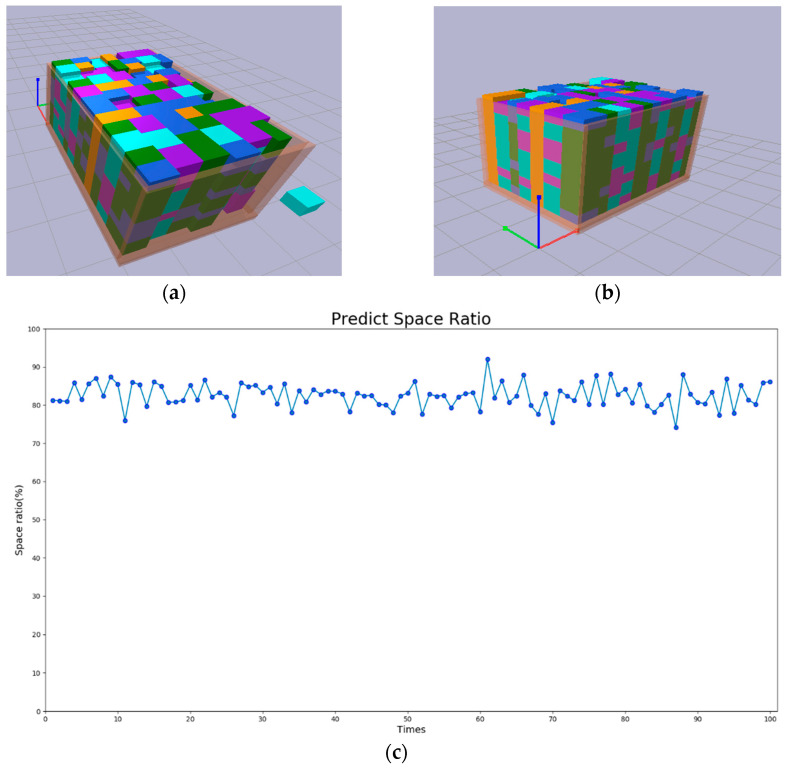
Test results of Model 1. (**a**) Top view of the packed result. (**b**) Bottom view of the packed result. (**c**) Space utilization rate for each round of the 100 tests. (**d**) Number of objects placed in each of the 100 tests.

**Figure 9 sensors-24-05370-f009:**
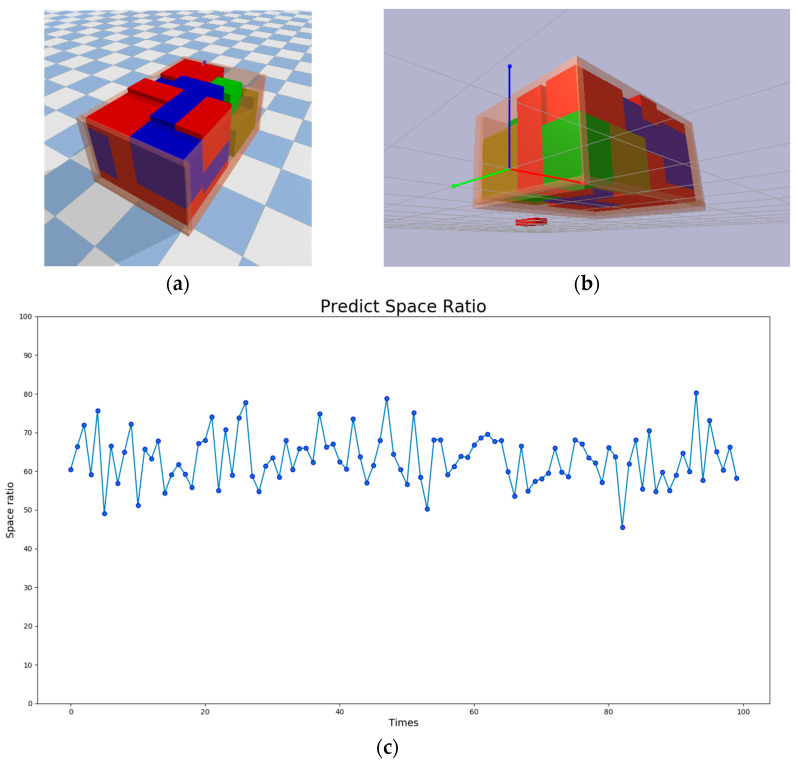
Test results of Model 2. (**a**) Top view of the packed result. (**b**) Bottom view of the packed result. (**c**) Space utilization rate for each round of the 100 tests. (**d**) Number of objects placed in each of the 100 tests.

**Figure 10 sensors-24-05370-f010:**
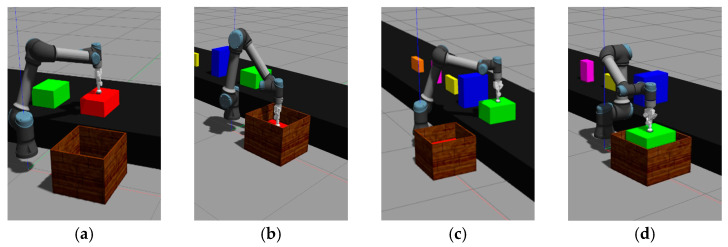
Snapshot of the online 3D-BP task simulation in Gazebo based on the proposed method. (**a**) The robot sucks the first object and approaches the packing bin. (**b**) The robot moves the first object towards the pre-determined position inside the bin. (**c**) The robot sucks the second object and approaches the packing bin. (**d**) The robot moves the second object towards the pre-determined position inside the bin. (**e**) The robot sucks the third object and approaches the packing bin. (**f**) The robot moves the third object towards the pre-determined position inside the bin. (**g**) The robot sucks the fourth object and approaches the packing bin. (**h**) The robot moves the fourth object towards the pre-determined position inside the bin. (**i**) The robot sucks the fifth object and approaches the packing bin. (**j**) The robot moves the fifth object towards the pre-determined position inside the bin. (**k**) The robot sucks the sixth object and approaches the packing bin. (**l**) The robot moves the sixth object towards the pre-determined position inside the bin.

**Figure 11 sensors-24-05370-f011:**
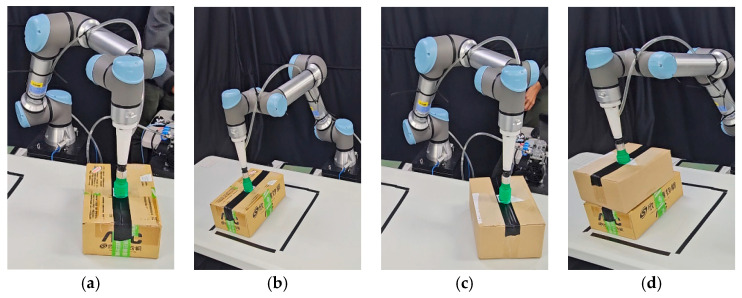
Snapshot of the online 3D-BP task with real robot manipulator based on the proposed method. (**a**) The robot sucks the first object and approaches the packing bin. (**b**) The robot moves the first object towards the pre-determined position inside the bin. (**c**) The robot sucks the second object and approaches the packing bin. (**d**) The robot moves the second object towards the pre-determined position inside the bin. (**e**) The robot sucks the third object and approaches the packing bin. (**f**) The robot moves the third object towards the pre-determined position inside the bin. (**g**) The robot sucks the fourth object and approaches the packing bin. (**h**) The robot moves the fourth object towards the pre-determined position inside the bin. (**i**) The robot sucks the fifth object and l approaches the packing bin. (**j**) The robot moves the fifth object towards the pre-determined position inside the bin. (**k**) The robot sucks the sixth object and approaches the packing bin. (**l**) The robot moves the sixth object towards the pre-determined position inside the bin.

**Figure 12 sensors-24-05370-f012:**
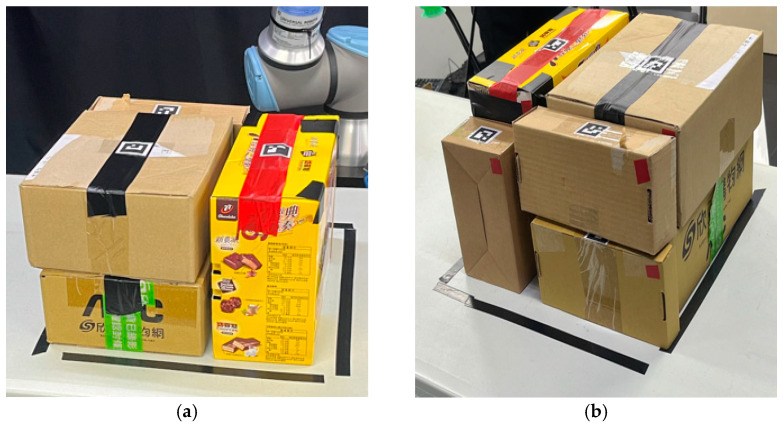
Results of an online 3D-BP task by the proposed method for a real robot manipulator. (**a**) Top-front view of the bin. (**b**) Side view of the bin.

**Table 1 sensors-24-05370-t001:** Parameter table of reward function.

Parameters	Definition	Value
σ	Standard for space utilization	80
β	Constant	0.2
γ	Constant	0.6
L	Length of container	400 (cm)
W	Width of container	300 (cm)
H	Height of container	200 (cm)
ρ	Constant	0.7
μ	Constant	0.3
τ	Constant	0.2
ω	Constant	0.8

**Table 2 sensors-24-05370-t002:** Object information of Model 1.

Object	Size (cm)	Color
Object 1	30 × 40 × 20	Orange
Object 2	30 × 50 × 20	Blue
Object 3	40 × 50 × 20	Purple
Object 4	30 × 50 × 40	Green
Object 5	40 × 50 × 30	Light blue
Container	400 × 300 × 200	Wood color (transparent)

**Table 3 sensors-24-05370-t003:** Space utilization in each round between research [31] and the proposed method.

Space Utilization	Research [31]	HHPPO	Comparison
Highest	85%	92%	Increase 7%
Top 5%	83.2%	89.2%	Increase 6%
Average	80%	83%	Increase 3%
Bottom 5%	72.4%	75.8%	Increase 3.4%
Lowest	70%	74%	Increase 4%

**Table 4 sensors-24-05370-t004:** Number of objects placed in each round between research [31] and the proposed method.

Space Utilization	Research [31]	HHPPO	Comparison
Highest	475	505	Increase 30
Top 5%	473	500	Increase 27
Average	440	455	Increase 15
Bottom 5%	410	424	Increase 14
Lowest	395	405	Increase 10

**Table 5 sensors-24-05370-t005:** Object information of Model 2.

Object	Size (cm)	Color
Object 1	50 × 100 × 20	Red
Object 2	30 × 90 × 10	Brown
Object 3	50 × 50 × 50	Blue
Object 4	60 × 60 × 10	Green
Container	300 × 200 × 150	Wood color (transparent)

**Table 6 sensors-24-05370-t006:** Object information of implement model.

Object	Size (cm)	Color
Object 1	21 × 28 × 10	Red
Object 2	21 × 25 × 12	Green
Object 3	12 × 25 × 23	Blue
Object 4	21 × 4 × 12	Yellow
Object 5	11 × 7 × 18	Pink
Object 6	22 × 6 × 11	Orange
Container	35 × 35 × 24	Wood color

## Data Availability

All data used to support the findings of this study are available from the corresponding author upon request.

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
