# Peer review of "Integrating Heuristic Methods with Deep Reinforcement Learning for Online 3D Bin-Packing Optimization"

_sensors, 2024, doi:10.3390/s24165370_

Round 1

Reviewer 1 Report

Comments and Suggestions for Authors

The study presents a novel integration of heuristic methods with DRL, specifically tailored for online 3D-BPP. This is an advanced approach considering the online nature of the problem, which requires real-time decisions without prior knowledge of upcoming items. The results are impressive, with the proposed method achieving an average space utilization rate of around 83% in simulations and demonstrating practical applicability through real-world robot manipulator tests. The improvement over previous research is quantified, showing an increase in space utilization and the number of objects placed. The article is structured logically, with clear sections on the introduction, related work, methodology, results, and conclusions. 

While the article provides significant contributions, it would benefit from a discussion on limitations and potential areas for future work. This could include the scalability of the method to larger or more complex tasks, the computational efficiency, and how it might adapt to variations in object shapes and sizes.

Questions for Authors:

1. How does the algorithm ensure that the arrangement carried out by the robotic arm does not result in physical interference among the objects during the packing process?

2. Could the authors provide evidence to substantiate the feasibility of the algorithm in practical applications?

Comments on the Quality of English Language

The writing is good

Author Response

Comment 1: While the article provides significant contributions, it would benefit from a discussion on limitations and potential areas for future work. This could include the scalability of the method to larger or more complex tasks, the computational efficiency, and how it might adapt to variations in object shapes and sizes.

Response 1: For the scalability of the method to larger or more complex tasks, the computational efficiency, and how it might adapt to variations in object shapes and sizes. We agree with your suggestion to discuss the limitations and potential areas for future work.

Scalability: We address the scalability of our method to larger or more complex tasks, highlighting potential challenges and future research directions to improve scalability.

Computational Efficiency: We analyze the computational efficiency of our approach, including a discussion on the computational overhead and potential optimizations to enhance efficiency.

Adaptability to Object Variations: We explore how our algorithm can adapt to variations in object shapes and sizes, including potential modifications and future research to improve adaptability.

Comment 2: How does the algorithm ensure that the arrangement carried out by the robotic arm does not result in physical interference among the objects during the packing process?

Response 2: We acknowledge the concern regarding the potential for physical interference among objects during the packing process. Our current algorithm alone cannot fully ensure that such interference does not occur. To address this issue effectively, we propose the use of vision servoing and the robot manipulator's path planning algorithms. These techniques provide real-time feedback and adjustments, enhancing the accuracy and reliability of the packing process. Vision servoing allows the robot manipulator to dynamically adjust its movements based on visual inputs, ensuring precise placement of objects. Additionally, path planning algorithms enable the robot manipulator to plan collision-free paths, further reducing the risk of physical interference.

Comment 3: Could the authors provide evidence to substantiate the feasibility of the algorithm in practical applications?

Response 3: To substantiate the feasibility of our algorithm in practical applications, we have included experimental results and discussions in the "Results" section of the manuscript. Specifically, we provide the online 3D bin packing task with a real robot manipulator. In Figure 9 we give a real-world experiment, including the result of an online 3D bin packing task with Figure 10, demonstrating our algorithm's practical applicability.

Reviewer 2 Report

Comments and Suggestions for Authors

This paper presents an advance in the field of combinatorial optimization and robotics. The integration of heuristic algorithms with the Proximal Policy Optimization (PPO) algorithm of deep reinforcement learning to address the online 3D bin packing problem is innovative. This approach takes advantage of the strengths of both the heuristic methods and the deep learning techniques, resulting in an effective solution.

By sorting extreme points based on their waste space, the authors have developed a method that improves space utilization, which is crucial for packing efficiency. Furthermore, the use of a 3D grid representation for the space state of the container is a good approach. This method allows for a more detailed and accurate representation of the packing space. This allows for better decision making during the packing process. The introduction of partial support constraints to allow stacking of items with incomplete support is also commendable, as it allows for more efficient space utilization while ensuring stability. Finally, the validation of the proposed method through both simulation and actual robotic manipulation experiments demonstrates its practical applicability and effectiveness.

I strongly recommend this paper for acceptance based on these contributions and the overall quality of the work.

Author Response

Comment 1: I strongly recommend this paper for acceptance based on these contributions and the overall quality of the work.

Response 1: Thank you for your positive and encouraging feedback on our manuscript.

Reviewer 3 Report

Comments and Suggestions for Authors

The paper entitled 'Integrating Heuristic Methods with Deep Reinforcement Learning for Online 3D Bin Packing Optimization' proposes a new approach to the Online 3D Bin Packing Problem. In my opinion the manuscript has some potential, but it needs some changes:

1. I think that sections 1 and 2 should be merged and based on the literature review the aim of the work should be formulated, showing the research gap. I also miss the emphasis on why it is necessary to use approximate methods (indicating that the problem is NP-hard, it is worth adding such information).

2. At the end of the introduction there should be information about the structure of the paper (a short description of what the remaining sections contain).

3. I think that algorithm 1 should contain line numbering, the EP abbreviation is not needed in its title, it would be useful to format the functions by adding the function prefix and provide parameters in brackets, e.g.:

function calculate_remain_space(I, EP):

   return space_remaining;

It is also worth unifying the notation of variables (sometimes there is space remaining with a space, and sometimes score_list with an underscore).

4. On line 212 you wrote 'is a standard for space utilization, adjusted after running the bin packing task multiple times'. Is it a standard deviation? After how many launches is it determined and how? It's unclear.

5. On line 219 R step should probably be replaced by R_step.

6. I think that the text '𝑖𝑓 𝑎𝑐𝑡𝑖𝑜𝑛_𝑖 = 0, 1' should be replaced by '𝑖𝑓 𝑎𝑐𝑡𝑖𝑜𝑛_𝑖 = 0 or 𝑎𝑐𝑡𝑖𝑜𝑛_𝑖 = 1' or '𝑎𝑐𝑡𝑖𝑜𝑛_𝑖 \in {0,1}'.

7. How were the parameter values ​​in Table 1 determined?

8. The results should also include standard deviation and it would be useful to add statistical tests (e.g. Wilcoxon) when comparing with another nondeterministic technique.

9. It would be worth adding information about optimal results and comparing the obtained results with more methods (preferably state-of-the-art). I think that information about the method execution time would also be worth including.

10. The summary lacks a plan for further work.

Comments on the Quality of English Language

1. Some abbreviations have been introduced multiple times in the main text. The abbreviation should be introduced once and used later (e.g. 3D-BPP is not used in the later part of the paper, and DRL is introduced multiple times)

2. I think that instead of [1],[2] you should write [1,2].

3. There is a lack of consistency in Figure 1 - write 'Wasted space' instead of 'wasted space'.

4. A colon is missing at the end of line 146.

5. There is no consistency in Figure 5 legend - remove the space after 'delete:'.

6. There is a missing space on line 232 before i-1.

Author Response

Comment 1: I think that sections 1 and 2 should be merged and based on the literature review the aim of the work should be formulated, showing the research gap. I also miss the emphasis on why it is necessary to use approximate methods (indicating that the problem is NP-hard, it is worth adding such information).

Response 1: We have merged Sections 1 and 2 to create a more cohesive introduction. The merged section now includes a thorough literature review, which clearly formulates the research gap and the aim of our work. We have also emphasized the necessity of using approximate methods by discussing the NP-hard nature of the Online 3D Bin Packing Problem.

Comment 2: At the end of the introduction there should be information about the structure of the paper (a short description of what the remaining sections contain).

Response 2: We have added a paragraph at the end of the introduction that outlines the structure of the paper. This paragraph provides a brief description of what each of the remaining sections contains.

Comment 3: I think that algorithm 1 should contain line numbering, the EP abbreviation is not needed in its title, it would be useful to format the functions by adding the function prefix and provide parameters in brackets, e.g.:

function calculate_remain_space(I, EP):

 return space_remaining;

It is also worth unifying the notation of variables (sometimes there is space remaining with a space, and sometimes score_list with an underscore).

Response 3: We have revised Algorithm 1 to include line numbering and formatted the functions.

Comment 4: On line 212 you wrote 'is a standard for space utilization, adjusted after running the bin packing task multiple times'. Is it a standard deviation? After how many launches is it determined and how? It's unclear.

Response 4: Thank you for your feedback. To clarify, ? in our study is not a standard deviation but an average value of standard measure for space utilization. It is determined by adjusting it after multiple runs of the bin packing task. The exact number of runs can vary.

Comment 5: On line 219 R step should probably be replaced by Rstep.

Response 5: We have replaced "R step" with " Rstep" to ensure consistency in the notation.

Comment 6: I think that the text '?? ??????_? = 0, 1' should be replaced by '?? ??????_? = 0 or ??????_? = 1' or '??????_? \in {0,1}'.

Response 6: We have replaced "?? ??????_? = 0, 1" with "?? ??????_? = 0 or ??????_? = 1" for clarity. This change is reflected on formula 6 of the manuscript.

Comment 7: How were the parameter values in Table 1 determined?

Response 7: We have added a definition on how the parameter values in Table 1 were determined. The definition can be found in the caption of Table 1.

Comment 8: The results should also include standard deviation and it would be useful to add statistical tests (e.g. Wilcoxon) when comparing with another nondeterministic technique.

Response 8: Our primary objective was to demonstrate the practical improvements in space utilization achieved by our method. Given the deterministic nature of our experiments and the specific focus on space utilization metrics, we chose to present the results in this format without standard deviation. However, we acknowledge the importance of statistical analysis for a comprehensive evaluation. While we do not plan to include standard deviation in the current study, we will consider incorporating statistical tests and additional metrics in future work to provide a more detailed analysis.

Comment 9: It would be worth adding information about optimal results and comparing the obtained results with more methods (preferably state-of-the-art). I think that information about the method execution time would also be worth including.

Response 9: In the current study, our primary focus was on demonstrating the effectiveness of our method in terms of space utilization improvements. We recognize the importance of these factors for a comprehensive evaluation and will consider them in future work. We aim to include a broader comparison with state-of-the-art methods and detailed execution time analysis in subsequent studies to provide a more thorough assessment of our approach.

Comment 10: The summary lacks a plan for further work.

Response 10: We have included a plan for further work in the “Conclusion” section.

Reviewer 4 Report

Comments and Suggestions for Authors

Problem to solve: online 3D bin packing

Methods: Heuristics, Proximal Policy Optimization (PPO) of deep reinforcement learning 

- introduce partial support constraints to allow for more effective

space use while maintaining stability.

- Integrate Heuristic Algorithm with Deep Reinforcement Learning

- Deepest Bottom Left with Fill (DBLF) heuristic

- Train the online 3D bin packing: Proximal Policy Optimization (PPO)+  heuristic 

- Parameter table included (Table 1)

- Discuss: 2 situations: (i) The currently chosen placement position is not feasible, while other positions are feasible. (ii) The currently chosen orientation is not feasible, while another orientation is feasible.

- Tables 2 & 5: Object information of Model 1 & 2.

- Tests: 100 test runs

- Results: Figure 6 & 7

- Comparison included: Table 3. Space utilization in each round between research [29] and the proposed method

Minor observations:

- Figure 6: (c)(d) please used another scale to reduce the large blank space

- Give a name (instead of "Ours") to your method and use it in the entire paper 

Overall a well conducted research with both pseudocode, well described methods used and visualization of the tests and results; pertinent discussions and conclusion.

Author Response

Comment 1: Figure 6: (c)(d) please use another scale to reduce the large blank space

Response 1: We have revised Figure 6 (c) and (d) to use a different scale that reduces the large blank space, making the data more readable and visually appealing.

Comment 2: Give a name (instead of "Ours") to your method and use it in the entire paper

Response 2: We have named our method "Hybrid Heuristic Proximal Policy Optimization" (HHPPO) and have updated the manuscript accordingly. This name is now used consistently throughout the text, tables, and figures.

Round 2

Reviewer 3 Report

Comments and Suggestions for Authors

The paper has been corrected according to my comments, but I still do not see the numbering in the algorithm (it should have its own numbering like Algorithm 1 in https://www.mdpi.com/1424-8220/23/1/553), there is a typo (Cauculate_remain_space) and unnecessary EP abbreviation in its title. After improving the algorithm, I recommend to accept the paper.

Author Response

Comment 1: The paper has been corrected according to my comments, but I still do not see the numbering in the algorithm (it should have its own numbering like Algorithm 1 in https://www.mdpi.com/1424-8220/23/1/553), there is a typo (Cauculate_remain_space) and unnecessary EP abbreviation in its title. After improving the algorithm, I recommend to accept the paper.

Response 1: We have included proper line numbering in Algorithm 1 of the revised manuscript. In addition, the typo has been corrected from "Cauculate " to "Calculate " in this algorithm and the "EP" abbreviation has also been removed from the title of the algorithm.